

# Mechanisms of the time-varying sea surface height and heat content trends in the eastern Nordic Seas

Sara Broomé[1], Léon Chafik[1], and Johan Nilsson[1]

[1]Department of Meteorology and Bolin Centre for Climate Research, Stockholm University, Stockholm, Sweden

**Correspondence:** S. Broomé (sara.broome@misu.su.se)

**Abstract.** The Nordic Seas is the main ocean conveyor of heat between the North Atlantic Ocean and the Arctic Ocean. Although the decadal variability of the Subpolar North Atlantic has been given significant attention lately, especially regarding the cooling trend since mid-2000s, less is known about the potential connection downstream in the northern basins. Using sea surface heights from satellite altimetry over the past 25 years (1993-2017), we find significant variability on multiyear-

to-decadal time scales in the Nordic Seas. In particular, the regional trends in sea surface height show signs of a slowdown since mid-2000s as compared to the rapid increase in the preceding decade since early 1990s. This change is most prominent in the Atlantic origin waters in the eastern Nordic Seas and is closely linked, as estimated from hydrography, to heat content. Furthermore, we formulate a simple heat budget for the eastern Nordic Seas to discuss the relative importance of local and remote sources of variability; advection of temperature anomalies in the Atlantic inflow is found to be the main mechanism. A

conceptual model of ocean heat convergence, with only upstream temperature measurements at the inflow to the Nordic Seas as input, is able to reproduce key aspects of the decadal variability of the Nordic Seas' heat content. Based on these results, we argue that there is a strong connection with the upstream Subpolar North Atlantic. However, although the shift in trends in the mid-2000s is coincident in the Nordic Seas and the Subpolar North Atlantic, the eastern Nordic Seas has not seen a reversal of trends but instead maintain elevated sea surface heights and heat content in the recent decade considered here.

# 1 Introduction

The Nordic Seas, a collective name for the Greenland–Iceland–Norwegian Seas, are the link between the Atlantic and the Arctic Oceans and are recognized to play an important role in the global climate system (Drange et al., 2005). Warm and saline waters of Atlantic origin cross the Greenland–Scotland Ridge (Fig. 1) and flow northward through the eastern part of the Nordic Seas before entering into the Arctic Ocean (Mauritzen, 1996; Orvik and Niiler, 2002; Skagseth et al., 2008; Furevik et al., 2007),

affecting the local sea ice and atmosphere on its way. The densest waters sustaining the lower limb of the Atlantic Meridional Overturning Circulation Chafik and Rossby (2019) are also produced in this region, by heat loss to the atmosphere, before flowing southward at depth across the Greenland–Scotland Ridge into the North Atlantic Ocean (Mauritzen, 1996; Hansen and Østerhus, 2000).

Since the Nordic Seas is a major source of heat for the Arctic Ocean it is important to understand the thermal variability and

the mechanisms behind it. The Nordic Seas experience intrinsic variability on many time scales (e.g. Siegismund et al., 2007;





Mork et al., 2014; Glessmer et al., 2014; Eldevik et al., 2009; Årthun et al., 2017; Shi et al., 2017). Carton et al. (2011) find several warm and cold events on multiyear timescales in an extensive 60 year hydrographic record. Segtnan et al. (2011) use reanalysis to examine the heat and freshwater budgets and find the largest water mass modifications to occur in the eastern part of the Nordic Seas. Asbjørnsen et al. (2018) use a consistent model framework to set up a closed heat and freshwater budget.

A common question that these studies, and many others, address is if the source of variability is local, by interaction with the atmosphere, or remote and advected into the Nordic Seas. Anomalies have been found to propagate from the North Atlantic over the Greenland–Scotland Ridge (Årthun and Eldevik, 2016; Furevik, 2000; Koszalka et al., 2013) and the Atlantic inflow is tightly linked to dynamics of the Subpolar Gyre (Hátún et al., 2005). Interestingly, the Subpolar North Atlantic have recently experienced strong decadal variability (Robson et al., 2016; Piecuch et al., 2017; Ruiz-Barradas et al., 2018) but the possible

impacts of this further north are not well established. The focus of this study is on recent decadal variability in the Atlantic water domain in the eastern parts of the Nordic Seas, with emphasis on the mecanisms behind the variability.

For a couple of decades now, we have been able to monitor sea level change and study key aspects of ocean dynamics using satellite altimetry. The dynamic sea surface height (SSH) retrieved from satellites carry information on ocean circulation, as it represents streamlines of the surface geostrophic currents, as well as sea level. The SSH reflects both steric height and

dynamic bottom pressure (Broomé and Nilsson, 2016); regional sea level change can be related to warming/cooling and freshening/salinification by air–sea fluxes but also due to redistribution of mass, heat and freshwater by time-varying ocean currents (Stammer et al., 2013). On time scales from days to months, local wind forcing and rapidly propagating waves are the main drivers of variability in sea level (Stammer, 1997). On longer timescales, multiyear/interannual-to-decadal, which are relevant to this study, the steric component of the SSH due to the integrated buoyancy of the water column instead becomes the main

driver of the variability in sea level (Richter and Maus, 2011). The time series of satellite altimetry is available since 1993 and is now becoming useful for studying recent decadal variability (Chafik et al., 2019).

This study aims to analyze the decadal variability of the Nordic Seas, using the altimetric time series of dynamic SSH combined with in situ data. More specifically, in section 3.1 we find that in addition to a general positive trend in sea level, the Nordic Seas has had a period of rapid increase in SSH followed by a period of stagnant SSH. This decadal variability is

concentrated in the eastern, Atlantic origin waters. We show that the decadal variability in SSH is linked to heat content and through a heat budget and conceptual model in section 3.2 we argue that the variations in temperature of the inflowing Atlantic water in the south is the main contributor to the variability. A strong connection to recent decadal variability in the subpolar gyre, discussed in section 3.3, further strengthens this idea, but also raises some questions about possible variations in the connection over time.

## 2 Data and method

### 2.1 Satellite altimetry (SSH)

This study has been conducted using satellite altimetry retrieved from Copernicus Marine Service Information. We use Absolute Dynamic Topography (ADT) which is the sea surface height above the geoid, i.e. the part of the SSH related to the ocean





circulation. The gradient of the ADT is directly proportional to the surface geostrophic current. The ADT has undergone several correction, calibration and homogenization processes, bringing data from several satellite missions together (Pujol et al., 2016). The ADT is distributed as daily fields on a regular 1/4° grid and has here been averaged into monthly fields from 1993 until 2017 and then deseasonalized.

## 2.2 Hydrography

We use the EN4.2.0 data set provided by the UK Met Office (Good, Simon A. and Martin, Matthew J. and Rayner, Nick A., 2013), with bias correction by Gouretski and Reseghetti (2010), on a 1°horizontal grid and 42 depth levels with higher resolution closer to the surface. Similarly to the ADT, we make time series of deseasonalized monthly means.

From the hydrographic data, the steric height ($\eta_S$) and a baroclinic volume transport function ($\psi$), proportional to the potential energy anomaly, can be calculated

$$\eta_S = \int_{-h_b}^{0} \Delta\rho^* \, dz \tag{1}$$

$$\psi = -\frac{g}{f} \int_{-h_b}^{0} z\Delta\rho^* \, dz, \tag{2}$$

where

$$\Delta\rho^* = [\rho(34.9, -1, z) - \rho(S, T, z)]/\rho(34.9, -1, 0), \tag{3}$$

is a non-dimensional density anomaly that measures the density deficit of the Atlantic Water layer relative to the deep water, $g$ the acceleration of gravity, and $f$ the Coriolis parameter. The transport stream function $\psi$ is the potential energy anomaly divided by $f$ and represents the vertically-integrated thermal-wind flow from $z = -h_b$ to the surface. Note that in a 1.5-layer model with an active upper layer with the depth $H$, the steric height and baroclinic transport (or potential energy) are closely related and given by (Nilsson et al., 2005)

$$\eta_S = \Delta\rho^* H, \tag{4}$$

$$\psi = \frac{g\Delta\rho^* H^2}{2f}. \tag{5}$$

To capture the dynamics and heat content of the waters of Atlantic origin that occupy the eastern Nordic Seas, integrations are done down to a depth level representative for the depth of the Atlantic Water (AW), in this case the EN4 depth level 657m (see e.g. Skagseth and Mork, 2012). The deep water below the AW is colder and the thermal expansion coefficient lower, thus the contribution to the steric height is supposedly lower. By limiting the integration to 657 m, we exclude the contribution from the deep water that does not experience the same variability as the AW and is not directly affected by the North Atlantic. Similar results are obtained also for integrations extending down to about 1000 m.





## 2.3 Air-sea heat flux

The net air-sea heat flux have been calculated from five different sources of surface fluxes. Three are from atmospheric re-analysis: ERA-Interim (Dee et al., 2011), NCEP (Kalnay et al., 1996) and JRA-55 (Kobayashi et al., 2015). The NOC surface flux (Berry and Kent, 2009, 2011) is calculated from observations of bulk atmospheric properties and J-OFURO (Tomita et al., 2019) is satellite-derived.

## 3 Results and discussion

### 3.1 Sea surface height trends and heat content

Over the last three decades, the dynamic sea surface height in the Nordic Seas has generally been rising. Figure 2 shows the linear trend in SSH from 1993 to 2017, which is positive almost everywhere and has a local maximum in the Atlantic Water (AW) in the Lofoten basin of over half a centimeter per year. In parallel, hydrographic observations show that the steric height and the potential energy anomaly (Eqs. 1, 2) have increased during the same period; in Fig. 2 is also the trend in potential energy, or equivalently baroclinic volume transport, which largely mirrors the trend in steric height (not shown). Most of the hydrographic trend is in the Atlantic origin sector of the Nordic Seas and the local maximum is, similarly to the SSH, located in the Lofoten Basin. The heat content of the AW has also increased and a maximum can again be identified in the Lofoten Basin (Skagseth and Mork, 2012; Mork et al., 2014; Shi et al., 2017). The trends in SSH, steric height ($\eta_S$) and baroclinic volume transport ($\psi$) differ in the shallow shelf regions, but the broad features in areas within the AW that are deeper than 500 m are similar. The pattern of these trends resembles that of the time-mean steric height and in turn, as the buoyancy of the AW is essentially uniform, the time-mean steric height roughly map the depth of the AW layer (see Broomé and Nilsson, 2016, Fig. 3). What this reasoning suggests is that the trend in SSH is to a first approximation caused by a uniform warming of the AW. This notion is also supported by Skagseth and Mork (2012).

The pattern seen in the trends of the SSH, steric height, and potential energy resemble the pattern of the time-mean steric height, but only to a lesser extent that of the time-mean SSH (see e.g. Broomé and Nilsson, 2016, Fig. 3). This indicates that the general warming of the AW during the period 1993 to 2017 also has entailed a gradual reorganisation of the circulation both at the surface and over the depth of the AW. The circulation in the AW domain consists of a current system of two branches (Orvik and Niiler, 2002); The Norwegian Atlantic Front Current (NwAFC) and the Norwegian Atlantic Slope Current (NwASC), see Fig. 1. Figure 2 reveals a strengthening of an anticyclonic flow anomaly in the Lofoten Basin, which tends to divert water south-east of the LB towards the outer NwAFC branch, flowing along the western limit of the Lofoten Basin. Thus, near the Lofoten Basin the trends in AW density serve to strengthen the outer NwAFC branch at the expense of the inner NwASC branch. This is expected to augment the mean flow heat transport that enters the Lofoten Basin from south (Dugstad et al., 2019). Potentially, this could also increase the residence time of the AW in the region as an increasing fraction of the AW tends to follow the NwAFC, taking a longer path along the western edge of the Lofoten Basin. Periods with long-term trends of AW



cooling and densification can be expected to show similar patterns of trends in SSH and baroclinic flow as seen in Fig. 2 but with the reversed sign.

### 3.1.1 Decadal variability

Analysis of the time series of satellite altimetry reveals that the positive trend is not constant in time. Fig. 3 shows the linear
trend in SSH for two decades, one from 1993 to 2002 and the other from 2004 to 2013. These two periods have very different patterns; the first period has a pronounced positive trend in the AW and also in the Greenland Sea while the second period has smaller amplitudes and no clear sign of trend in the AW. It is clear that most of the linear increase seen in Fig. 2 occurs in the first of these periods.

It is also apparent that the greatest change in trend between the two decades occurs in the Atlantic water domain south of the
Barents Sea opening (BSO). To identify the Atlantic water variability we define the AW area, shaded gray in Fig. 1, as the area between 63.5 and 72.5°N that is enclosed by the time-mean position of the 35.0 surface isohaline. The defined area comprises the AW from the southern section where different inflows over the Greenland–Scotland Ridge merge into the eastern boundary current system, up to and including the deep pool of AW in the Lofoten Basin. North of this, the AW fractionates between the Barents Sea and the continental slope towards the Fram Strait.

The SSH is averaged over the defined AW area, resulting in the time series shown in Fig. 4. The data have been deseasonalized to remove the otherwise dominant seasonal cycle of high sea surface height in summer and low in winter, reflecting the seasonal variation in heat content. A large monthly variability remains in addition to a long term trend of about 0.3 cm yr$^{-1}$. Around 2004 or 2005, the area averaged SSH shifts from a period of high variability and positive trend to a stagnant period of smaller variability and no trend.

The variations in the SSH on multi-year and longer time scales is closely related to heat content (Richter and Maus, 2011; Shi et al., 2017) and Fig. 4 shows a corresponding AW time series of heat content. Although the correlation between the monthly time series of SSH and heat content is low, there is a definite similitude in the decadal trends; in Fig. 4 there seems to first be a period of strong positive trend of about 5 Wm$^{-2}$ followed by a stagnant period, with the shift in the mid of the 2000s. This indicates that the observed decadal variability in the SSH is mainly a steric signal, a conclusion made also by Shi et al. (2017).

The selection of the two periods is somewhat arbitrary, and the trends are generally sensitive to the endpoints. Therefore, the periods should only be considered as guidelines and the full time series is included for transparency. Here, the altimetric time series has been the basis for the choice. There is an anomalous high event in the SSH around 2003 in Fig. 4, which is well correlated with a deepening of the AW layer, mostly in the Lofoten Basin, and less with temperature or heat content (Skagseth and Mork, 2012). We have therefore chosen to exclude the year 2003 from the periods.

In the next sections we will discuss the mechanisms behind the changes in trends between the two decades. First, a heat budget will be set up to discern the relative influence of ocean advection and air-sea heat fluxes. Second, based on the heat budget, we will discuss a simple conceptual model to show that changes of the temperature of the Atlantic origin inflow is a likely source of decadal variability. Third, we will analyse the connection between the AW in the Nordic Seas and the upstream subpolar North Atlantic.



## 3.2 Simple heat budget

A simple heat budget might give insight to the causes of the decadal variability. We consider the heat budget for a fixed volume of Atlantic Water in the Nordic Seas ($V_{AW}$), defined by the lateral boundaries in Fig. 1 and down to a fixed depth representative of the depth of the AW. The heat content is defined by

$$H \overset{\text{def}}{=} c_p \int_{V_{AW}} T \, dV, \tag{6}$$

where $c_p$ is the heat capacity per unit volume for sea water and $T$ the temperature, and the heat budget is

$$\frac{dH}{dt} = C - Q. \tag{7}$$

Here, $C$ is the ocean heat convergence and $Q$ the upward net heat flux at the sea surface.

Let us now use the heat budget to examine the heat content for the two decadal periods of interest here, called 1 (first) and 2

(second). Subtracting the budget in Eq. (7) for each period, we can write

$$\frac{1}{A} < \frac{dH}{dt} >_1 - \frac{1}{A} < \frac{dH}{dt} >_2 = $$
$$\frac{< C >_1 - < C >_2}{A} - \frac{< Q >_1 - < Q >_2}{A},$$

where $<>_{1,2}$ are the averages over period 1 and 2, respectively, and $A$ the surface area of the AW domain. The heat content of the AW volume (Fig. 4) has a linear increase during the first period per unit area of about 5 W m$^{-2}$ and about 0 W m$^{-2}$ during

the second period, i.e.

$$\frac{1}{A} < \frac{dH}{dt} >_1 - \frac{1}{A} < \frac{dH}{dt} >_2 \approx 5 \text{ W m}^{-2}. \tag{8}$$

To analyze if the surface heat flux $Q$ can explain the observed decadal variability, we use observations of net air-sea heat flux. However, the estimates available of surface heat flux differ significantly in pattern, variability and mean state (see e.g. Carton et al., 2018). To demonstrate this, we use five different estimates of the net flux, defined positive upwards, and average over the

two periods of interest and over the AW area, see Table 1 and Fig. 5. In an annual mean, the whole AW area loses heat to the atmosphere but the mean heat loss in Table 1 varies about 20 W/m$^2$, or 25-30%, between the products. To explain the observed variability, assuming in turn that the ocean heat flux divergence is zero, the second decade would have to experience a higher heat loss to the atmosphere, i.e. $-(< Q >_1 - < Q >_2)/A > 0$. This is true for one of the surface flux products (NOC), while the other estimates are close to zero or almost 10 W/m$^2$ in the other direction. The spatial patterns of the difference in surface

heat flux between the two periods (Fig. 5) also vary significantly between the data sets, and none of these patterns match the SSH trend pattern (Fig. 2) with its distinct peak in the Lofoten Basin.

Several observational studies have found that the surface heat flux can only explain a smaller part of the low-frequency variations of the AW heat content in the Nordic Seas (Carton et al., 2011; Skagseth and Mork, 2012; Shi et al., 2017). A study of a physically consistent ocean state estimate also show that surface heat flux is not the main source of AW heat content





interannual variability (Asbjørnsen et al., 2018). Although the surface heat flux data is not conclusive, we argue that the surface heat flux is not the main source of the change in decadal trends. We will thus continue by considering the other possible source in our heat budget, namely the ocean heat convergence. In the next section we will quantify the convergence and try to disentangle the contribution from variations in temperature and transport respectively.

### 3.2.1 Conceptual model of ocean heat convergence

We will now show that temperature variations of the AW, flowing across the Greenland–Scotland Ridge and into the southern border of the Nordic Seas AW domain, can explain a significant fraction of the observed heat content variability. To demonstrate this, we model ocean heat convergence as

$$C = c_p \Delta T \Psi, \qquad \Delta T \stackrel{\text{def}}{=} T_i(t) - T_0(t), \tag{9}$$

where $T_i/T_0$ is the temperature of the in/outflowing AW and $\Psi$ the volume transport. Using this, the heat budget in Eq. (7) becomes

$$\frac{dH}{dt} = c_p \Delta T \, \Psi - Q. \tag{10}$$

We write the variables in the heat budget as a sum of a time-mean part (overbar) and a time varying part (prime): $\Delta T = \overline{\Delta T} + \Delta T' \, (= \overline{\Delta T} + T_i' - T_o')$, and similarly for $\Psi$ and $Q$. We choose $\overline{\Delta T}$, $\overline{\Psi}$ and $\overline{Q}$ so that

$$c_p \overline{\Delta T} \, \overline{\Psi} = \overline{Q} \tag{11}$$

This implies that the time-mean heat budget is $\overline{C} = \overline{Q}$, i.e. in the time-mean the heat convergence is balanced by the upward surface heat flux. The linearised Eq. (10), neglecting $\Delta T' \Psi'$, then becomes

$$\frac{dH}{dt} = c_p \Delta T' \overline{\Psi} + c_p \overline{\Delta T} \Psi' - Q'. \tag{12}$$

Further, we set

$$\frac{dH}{dt} = c_p V_{AW} \frac{dT}{dt} = c_p V_{AW} \frac{dT'}{dt} \tag{13}$$

where $T'$ is the mean AW temperature anomaly. We will now make two simplifying assumptions. First, since the center of mass of the AW is located near the Lofoten Basin, close to our AW domain's northern boundary, we assume that the outflow temperature $T_o'(t)$ is approximately equal to the mean AW temperature anomaly $T'(t)$. Second, based on the discussion of the surface heat flux in section 3.2 we will here assume that $Q'$ is small and thereby limit the analysis to the ocean heat convergence. Using these two simplifications in Eqs. (12) and (13), we obtain the following equation for the mean AW temperature anomaly:

$$\tau \frac{dT'}{dt} + T' = T_i'(t) + \overline{\Delta T} \Psi'(t)/\overline{\Psi} \qquad \tau \stackrel{\text{def}}{=} \frac{V_{AW}}{\overline{\Psi}}. \tag{14}$$





Here, the terms on the right-hand side are the forcings due to anomalies in inflow temperature and AW volume transport, respectively, and $\tau \sim$ 3–4 years is the residence time of the AW in the domain (Koszalka et al., 2013; Broomé and Nilsson, 2018).

Equation (14) is based on the reasonable assumption that the low-frequency ocean heat convergence is dominated by changes
of the AW circulation. To examine if variations in temperature or transport dominate the variation in heat convergence, we note that the ratio between the second and first term on the right-hand side of Eq. (14) is

$$\frac{\Psi'}{\overline{\Psi}} \left( \frac{T_i'}{\overline{\Delta T}} \right)^{-1}. \tag{15}$$

This is the ratio between the two driving terms and if it is small, temperature anomalies dominate over transport anomalies in the ocean heat convergence, and conversely when the ratio is large. The Svinøy Section is roughly located at the upstream
border of our Atlantic Water domain. Here, the mean AW transport is $\overline{\Psi} \sim 5$ Sv (Mork and Skagseth, 2010) and $\overline{\Delta T}$ can be estimated from the steady state heat budget (Eq. (10)) as $\overline{\Delta T} \approx \overline{Q}/(c_p \overline{\Psi})$; taking $\overline{Q}/A \sim 80$ W m$^{-2}$ (see table 1) gives $\overline{\Delta T} \sim 2$ °C. Further, measurements in the Svinøy Section indicate low-frequency flow and temperature anomalies (> 5 years) that give $\Psi'/\overline{\Psi} \sim 0.3$ and $T_i'/\overline{\Delta T} \sim 0.4$ (estimated from Fig. 7 in Mork and Skagseth, 2010). This gives a value of about 0.7 for the ratio in Eq. (15), suggesting that variations in temperature are slightly more important than variations in volume flow. We obtain
similar results using observations from the Faroe–Shetland Channel (Berx et al., 2013). We note that per unit area in the AW domain, the $c_p \Delta T' \overline{\Psi}$ term in Eq. (12) gives a heat convergence of 40 W m$^{-2}$ for a $\Delta T'$ anomaly of 1°C. Thus, a difference in inflow and outflow temperatures less than 0.5°C could explain the observed increase in heat content from the mid 1990s to around 2004.

Our considerations show that AW temperature variations can be more important for the ocean heat convergence than vari-
ations in AW volume transport. However, the ocean heat transport variations in sections across the AW, such as the Svinøy Section, tend to be dominated by variations in the volume transport (Asbjørnsen et al., 2018). The reason is that there is a net volume transport across the sections, which requires the heat transport to be defined relative to a reference temperature. This reference temperature, characterising a return flow, is usually taken to be 0 °C for AW heat transport in the Nordic Seas (Asbjørnsen et al., 2018). Using a reference temperature of 0 °C to estimate the heat transport anomaly in Eq. (12) gives an ef-
fective temperature difference $\overline{\Delta T} \sim 6$°C (the mean temperature of the section in °C), rather than $\overline{\Delta T} \sim 2$°C (the temperature difference between in- and out-flowing water) as used here for estimating the ocean heat convergence.

We also note that observations of volume transport at the southern inflows to the Nordic Seas show no indication of decadal trends over the time period (Berx et al., 2013; Østerhus et al., 2019; Hansen et al., 2015). The volume transport can also be estimated from the slope of SSH (Chafik et al., 2015). Over the slope in the FSC, such a barotropic calculation (not shown)
gives a mean transport of just under 4 Sv (consistent with the 4.1 Sv direct estimates by Rossby and Flagg (2012)), with monthly estimates ranging from 0 to 8 SV, but no decadal trends to explain the ones in the Nordic Seas, further supporting the notion that temperature variations dominate the heat convergence.

Motivated by the considerations of the heat budget, we examine if the simple model defined by Eq. (14) can reproduce the evolution of the mean AW temperature anomaly, given the inflow temperature as the only forcing. For this purpose, we estimate





the AW inflow temperature $T_i'(t)$ from sea surface EN4 temperatures in the Faroe–Shetland Channel and in the Svinøy Section from 1955 up to the present and integrate Eq. (14) numerically forward in time. In the calculations, we use the e-folding timescales ($\tau$) 2.5, 3.0, and 4.0 years, which are in the range of estimated residence times in the presently defined AW domain (Koszalka et al., 2013; Broomé and Nilsson, 2018). This range of $\tau$ implies that the temperature/heat-content anomaly evolution

in the model is influenced by the upstream temperature history a couple of years back in time. The heat content is related to the temperature by Eq. (13), using a mean depth of the AW layer of 700 m.

Figure 6 shows the proxies for the AW inflow temperature $T_i'(t)$ and the resulting modelled heat content anomaly as well as the low-pass filtered AW heat content anomaly estimated from the EN4 data (from Fig. 4). For easy comparison, the heat content anomalies have been set to zero in 1993. The results based on the Svinøy Section temperature (results are similar

when the Faroe–Shetland Channel temperature is used) show that the simple model reproduces the main features of the low-frequency evolution of the AW heat anomaly. For all three values of $\tau$, the modelled heat content anomaly increases from the mid 1990:s to the mid 2000:s with a more stagnant period following, in broad agreement with the observations. In effect, the model acts as a low-pass filter on the driving inflow temperature $T_i'(t)$, suppressing variability with time-scales shorter than about 4 years. Thus, the model heat anomaly is mainly forced by the general increase of the inflow temperature up to around

2003 and the more constant inflow temperature in the period thereafter. There are some obvious differences between the model and the observational estimate of the heat content. Specifically, the increase of the heat anomaly in the data from around 1997 to 2004 is weaker and possibly delayed a couple of years in the model. A strengthening of the AW volume transport with some 10–20 % or a decreased surface heat loss during this period may explain the deviation between the simple model and the data-based estimate of the heat content anomaly. In any case, this calculation shows that the variations of the inflow temperature at

the southern boundary are important for the observed low-frequency heat content variability in the AW domain.

## 3.3   Connection to the upstream subpolar North Atlantic

Since our results suggest that local air-sea heat fluxes cannot explain the decadal heat content variability and that ocean advection of temperature anomalies from the south is the main cause, it is reasonable to assume a close connection to the Subpolar North Atlantic (SPNA). In this regard, several studies have documented a link between SPNA temperature variability and the

Nordic Seas, mediated by advection of temperature anomalies along the eastern branch of the North Atlantic Current (Chepurin and Carton, 2012; Årthun and Eldevik, 2016; Årthun et al., 2017; Langehaug et al., 2019). We will now consider if such an advective connection can explain Nordic Seas AW temperature variations from the period from around 1993 to 2016.

Figure 7, based on Empirical Orthogonal Function analysis (Hannachi et al., 2007), shows that the leading mode of temperature variability in the SPNA (explained variance is 70%), of both surface and subsurface temperatures down to $\sim$400 m,

is dominated by pronounced decadal variability. However, while the decadal temperature changes in the eastern Nordic Seas (Fig. 7, left panel) track those in the SPNA (Fig. 7, right panel) during the 1993-2004 period, a clear disconnection is seen after 2005. Consistent with the SSH and heat content analysis (e.g. Fig. 4), Fig. 7 shows that while the SPNA has been cooling since the mid 2000s, the Nordic Seas has not. This disconnection suggests a weak relationship of the eastern Nordic Seas to changes





in the SPNA during its cooling phase (∼mid-2000s to 2016), but a strong connection, as shown here and documented by many studies (Hátún et al., 2005; Skagseth and Mork, 2012), during the warming phase of the SPNA (∼early 1990s to mid-2000s).

This weakened (enhanced) connection between the SPNA and the Nordic Seas during the recent cooling (warming) phase may be explained by the horizontal circulation and hence the shape of the subpolar gyre/front in the eastern SPNA. As the subpolar gyre strengthens (weakens) during the cooling (warming) phase, in response to several years of strong (weak) wind-stress curl (Häkkinen et al., 2011), it also expands (contracts) in size and the subpolar front shifts eastwards (westwards), which, in turn, leads to a smaller (larger) fraction of subtropical water masses spreading along the eastern boundary, across the Greenland–Scotland Ridge and into the Nordic Seas. This view is supported by a spatial correlation analysis between the leading mode of temperature variability in the Nordic Seas at 100 m against that in the wider North Atlantic, shown in Fig. 8. The resulting pattern indicates that the relationship with the SPNA is only strong and significant along the subtropical path in the eastern subpolar gyre and around its rim rather than in the central SPNA, where the correlation is weak, negative and not significant. It is thus possible that the contraction and expansion of the subpolar gyre, through its control on the northward access of warm and saline subtropical waters in the eastern subpolar gyre (Hátún et al., 2005; Häkkinen et al., 2011; Chafik et al., 2019), may have regulated the observed time-varying connection between the SPNA and the Nordic Seas (Figs. 4 and 7) and hence the rate of ocean heat content and sea surface height change in the eastern Nordic Seas on decadal scales.

The subpolar gyre linkage discussed above together with the simple model in section 3.2.1, which only includes inflow temperature variations, suggest that air-sea fluxes could not have caused the observed shift in decadal heat content trends. However, this does not rule out that atmospheric circulation anomalies may also have helped to maintain warm ocean temperatures in the Nordic Seas since mid-2000s, resulting in the stagnant period instead of the cooling seen in the SPNA. This may be consistent with conclusions from a recent study by Asbjørnsen et al. (2018), which reported that local forcing from air-sea heat fluxes are important for modulating the anomalies on their northward path. Although we have only speculated on the cause for this stagnant period, our simple model of heat convergence reproduced the decadal heat content variability in the eastern Nordic Seas reasonably well, despite the observed disconnection from the SPNA. This important result thus suggests that temperature conditions in the northeastern Atlantic and at the Greenland–Scotland Ridge are key in setting the decadal variability in ocean heat content and SSH changes in the eastern Nordic Seas.

## 4   Summary and concluding remarks

In this study of the Atlantic water of the Nordic Seas (AW), decadal changes in sea surface height (SSH) since 1993 have been analysed and underlying mechanisms have been investigated. Over the full period of satellite observations of SSH, 1993-2017, there is a general positive trend coincident with a warming of AW in the eastern Nordic Seas. We identify a shift in the trend of SSH in the AW, from a decade of strong positive trend to a more stagnant decade. A similar change in trend is also found in the heat content of the AW. We argue that the steric height changes related to the variation in heat content is the main reason for the observed decadal changes in SSH trends.





Through a simple heat budget adapted to the AW, we discuss three possible reasons for the decadal variations in heat content: a difference in net surface heat flux between the two periods and a difference in either volume transport or temperature of the AW inflow at the southern boundary. We conclude that the most plausible cause of changes in SSH and heat content decadal trends is a change of temperature of the Atlantic source waters entering the Nordic Seas over the Greenland–Scotland Ridge.

A quantitative estimate of the relative contributions from volume transport and temperature in the heat transport shows that a difference in inflow temperature of $0.5°C$ is enough to explain the decadal changes in heat content. Furthermore, we construct a simplified conceptual heat budget model to forecast the AW heat content based on a single measurement of temperature at one of the main inflow regions to the Nordic Seas, i.e. the Faroe-Shetland Channel. The model is able to reproduce the main features of the observed decadal changes of AW heat content and SSH in time and magnitude.

Our main findings include:

- The decadal variability in SSH is closely related to heat content and the main reason for the shift in decadal trends in the SSH is the steric height changes related to heat content.

- The source of the decadal changes in heat content is remote, and most likely associated with a change of temperature of the Atlantic source waters entering the Nordic Seas over the Greenland–Scotland Ridge.

- A conceptual heat budget model of the AW heat content, based on a single measurement of inflow temperature, is able to roughly capture the observed decadal changes in the AW.

- The AW in the Nordic Seas has not experienced the same reversal of trends as the central Subpolar North Atlantic, but instead seem more related to the rim of the SPNA and maintain warm ocean temperatures and high SSH during the period considered.

The AW center of mass in the eastern Nordic Seas is encountered in the Lofoten Basin, where the AW layer extends deeper (Skagseth and Mork, 2012; Raj et al., 2015). Here the AW warming during the period 1993-2017 is most pronounced, which via associated changes in steric height and potential energy (Fig. 2) have served to induce an anticyclonic flow anomaly carrying a larger fraction of AW from the slope current into the Lofoten Basin. This flow anomaly acts to enhance the near-surface heat transport by the mean flow entering the Lofoten Basin from south (Dugstad et al., 2019). In combination with alterations of

eddy fluxes from the Lofoten escarpment (Spall, 2010; Chafik et al., 2015; Dugstad et al., 2019) the anticyclonic mean-flow anomaly are plausible mechanisms for the build-up of the Lofoten Basin heat content over the period 1993-2017.

We have connected decadal variability to upstream conditions, which has implications for decadal climate predictability in the Nordic Seas. It is equally important to note that the conditions observed in the AW of the Nordic Seas can have impact further downstream (Polyakov et al., 2017; Smedsrud et al., 2013; Sandø et al., 2010). For example, discussions in the past

few years have referred to an "Atlantification" of the Barents Sea and Arctic Ocean as a cause of sea ice loss (see e.g. Årthun et al., 2012; Polyakov et al., 2017), highlighting the importance of upstream AW conditions. Last but not least, this study shows that the satellite absolute dynamic topography can be used to study decadal variability in heat content, which is useful since





the satellite data is accessible, has no summer bias and generally has better and more even resolution in time and space than hydrography.

*Data availability.* The absolute dynamic topography data are available from http://marine.copernicus.eu/services-portfolio/access-to-products/ with the product identifier: SEALEVEL_GLO_PHY_L4_REP_OBSERVATIONS_008_047.

5   The EN4 hydrographic data are available from https://www.metoffice.gov.uk/hadobs/en4/.

The ERA-Interim, NCEP/NCAR and JRA reanalysis along with the NOC surface flux are available from the Research Data Archive: https://rda.ucar.edu/. The J-OFURO data are available from https://j-ofuro.scc.u-tokai.ac.jp/en/.

*Author contributions.* SB carried out the analysis from an original idea of LC and with continuous input from LC and JN. SB wrote the bulk of the paper, with JN as main author of section 3.2 and LC of section 3.3.

10  *Competing interests.* The authors declare no competing interests.

*Acknowledgements.* This work was supported by grants from the Swedish National Space Agency (Dnr 133/17, 111/16). We thank Jonas Nycander for valuable comments.





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



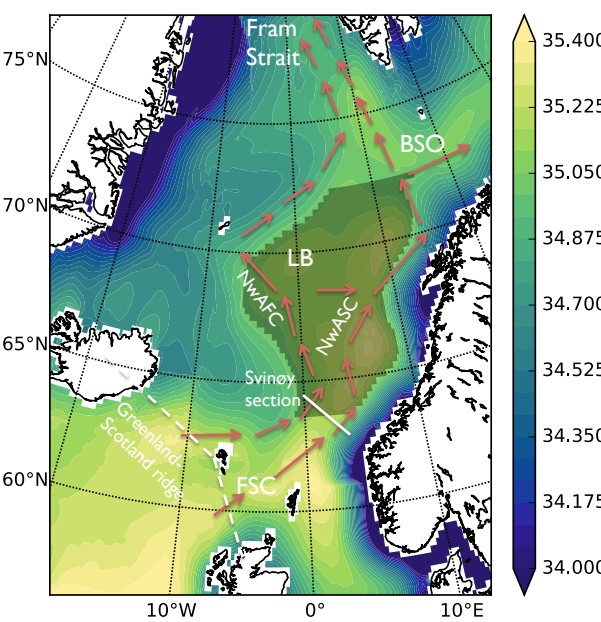

**Figure 1.** Map of the Nordic Seas with surface salinity (Zweng et al., 2013) in shading. The general pathways of the Norwegian Atlantic Front Current (NwAFC) and the Norwegian Atlantic Slope Current (NwASC) are indicated together with the Lofoten Basin (LB), Barents Sea Opening (BSO), Fram Strait, parts of the Greenland-Scotland ridge, the Faroe-Shetland channel (FSC) and the Svinøy hydrographic section. The darker shaded region is the area enclosed by the 35 surface salinity isohaline between 63.5 and 72.5°N. This area represents the Atlantic Water area in the current study.





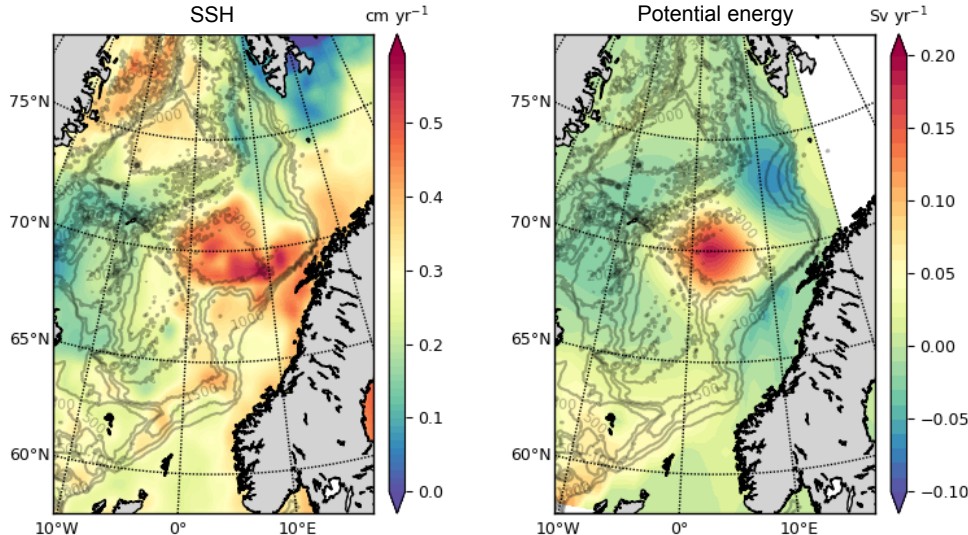

**Figure 2.** Left panel: Linear trend in SSH [cm yr$^{-1}$] over the full altimetry period 1993-2017. Right panel: Linear trend in potential energy [Sv yr$^{-1}$] (see Eq. (2)) from 1993-2016. Gray contours are bathymetry (Becker et al., 2009).



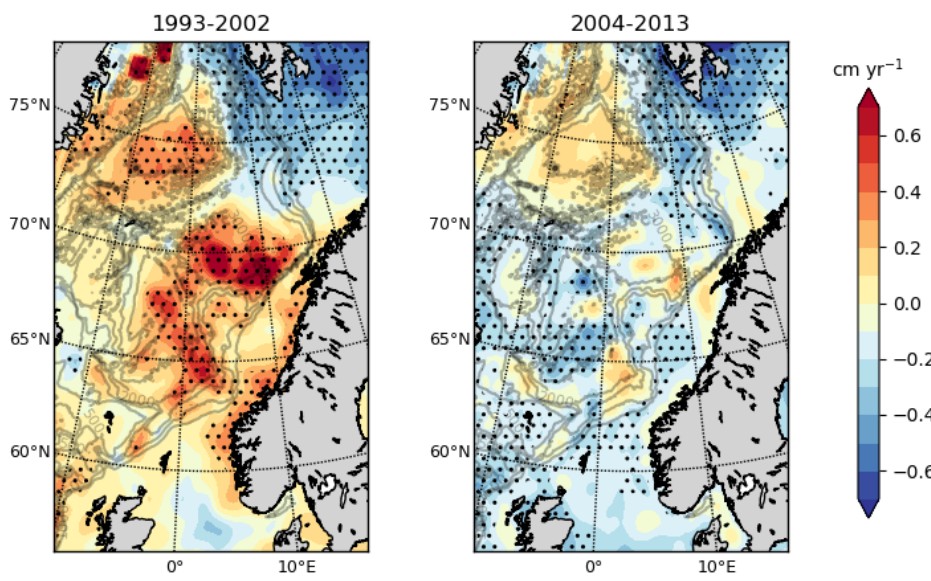

**Figure 3.** Linear trend in SSH [cm yr$^{-1}$] for (left) 1993-2002 and (right) 2004-2013. A global trend for the full altimetry period 1993-2017 has been removed from the data. Dotted areas are significant at a 95% confidence level. Gray contours are bathymetry (Becker et al., 2009).



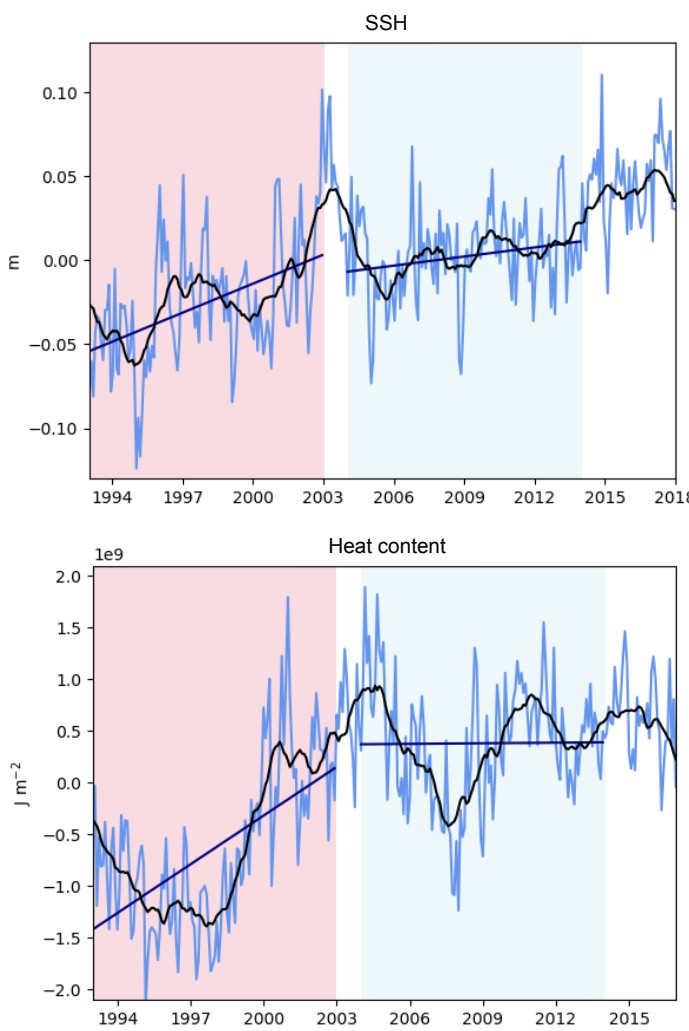

**Figure 4.** Upper panel: Monthly (light blue) and lowpass filtered (black) SSH [m] from 1993-2017 averaged over the AW area (see fig 1). The SSH has been deseasonalized. In dark blue are the linear trends of the SSH for the two periods 1993-2002 and 2004-2013. Lower panel: Monthly (light blue) and lowpass filtered (black) heat content in the upper 657 m of the AW area [J m$^{-2}$]. The heat content has been deseasonalized. In dark blue are the linear trends for the two periods 1993-2002 and 2004-2013.



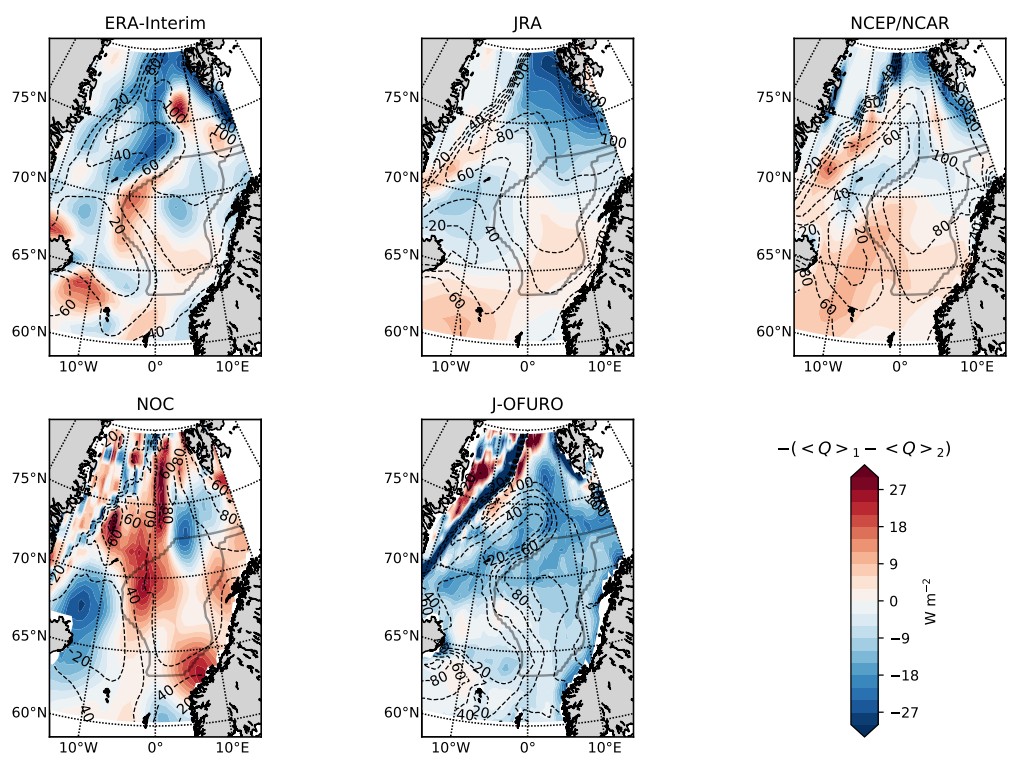

**Figure 5.** Net surface heat flux $Q$ [W m$^{-2}$], defined positive upwards, for five different products (ERA-Interim, JRA-55, NCEP/NCAR, NOC Surface Flux Dataset, J-OFURO). Shading represents the difference between the periods 1993-2002 and 2004-2013, i.e. $-(<Q>_1 - <Q>_2)$, overlaid by the climatological mean net surface heat flux 1993-2013 (dashed contours). The solid gray line depicts the AW domain, cf. Fig. 1.



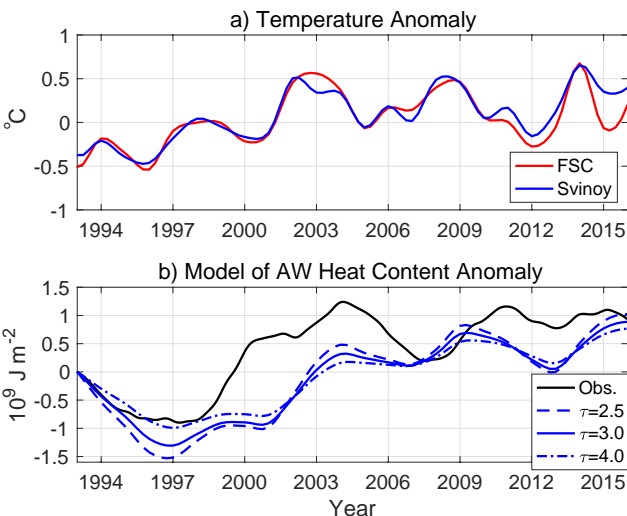

**Figure 6.** a) Sea surface EN4 temperature anomaly [°C] in the Faroe–Shetland Channel (FSC, red) and in the Svinøy Section (Svinoy, blue), used as proxies for the inflow temperature anomaly $T_i'(t)$. b) Modelled AW heat content anomaly (blue), which is obtained by integrating Eq. (14) forward in time, using the Svinøy Section temperatures and the e-folding timescales ($\tau$) equal to 2.5, 3.0, and 4.0 years. The black line shows the heat content AW anomaly estimated from the EN4 data; same data shown in Fig. 4 but here low passed with a 24 month running mean.





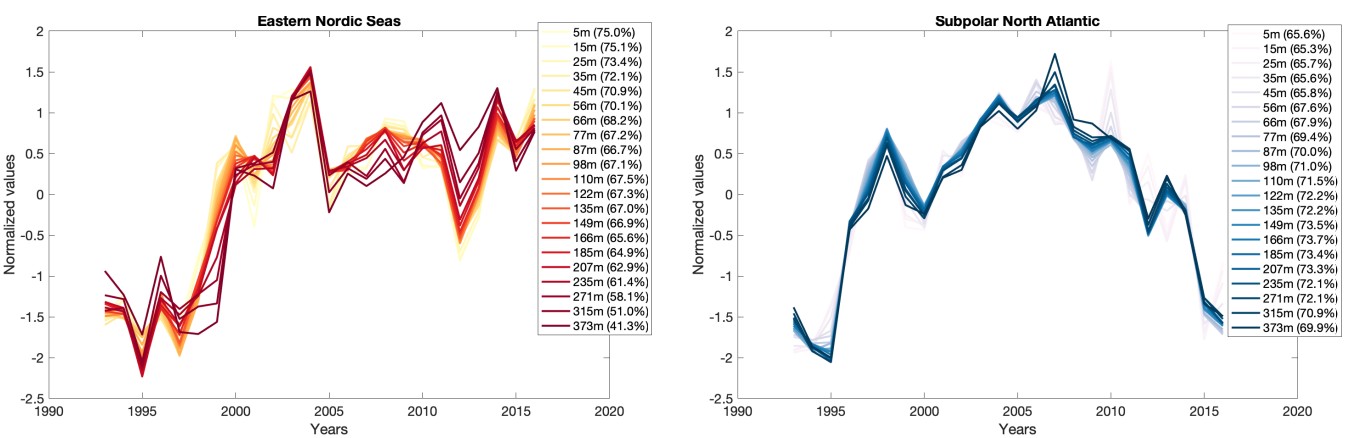

**Figure 7.** The Leading mode of variability of the deseasonalized (no trend removed) and annually-averaged temperature anomalies down to 400 m in the Eastern Nordic Seas (10°W-15°E,62-72°N, left panel) and the Subpolar North Atlantic (45-5°W, 55-65°N, right panel) for the 1993-2016 period using EN4. The explained variance of every depth can be seen in parentheses in the figure legend.



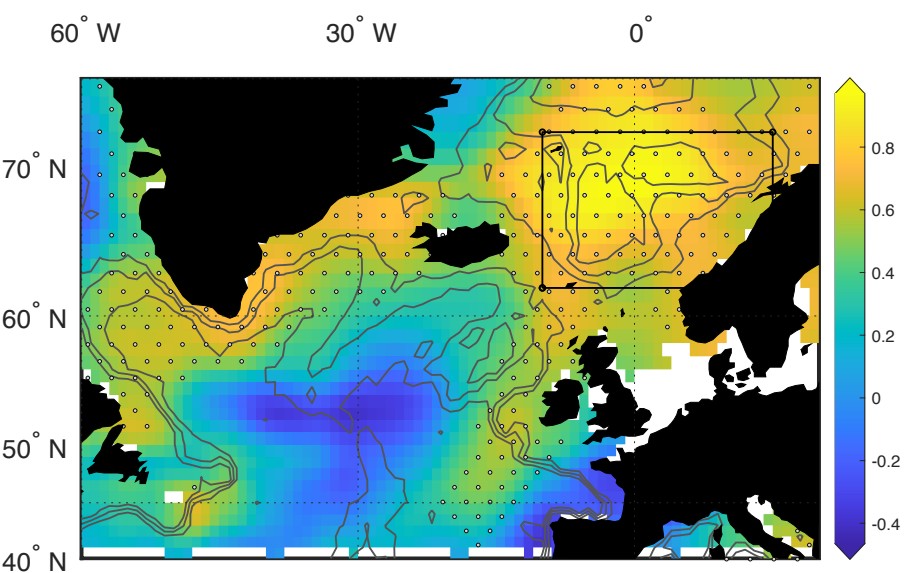

**Figure 8.** Correlation analysis between the Leading mode of variability of the deseasonalized (no trend removed) and annually-averaged temperature anomalies at 100 m in the eastern Nordic Seas (cf. left panel of Fig. 7) against every grid point of that in the North Atlantic using EN4. The black box encloses the area for the eastern Nordic Seas used here. Circles indicate significance at the 95% confidence level. Gray contours are bathymetry.



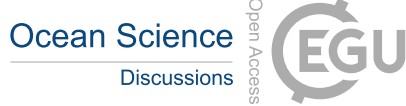

**Table 1.** Net surface heat flux $Q$ [W m$^{-2}$], defined positive upwards, averaged over the AW area (defined in Fig. 1) for the first period 1993-2002 ($<>_1$) and the second period 2004-2013 ($<>_2$). The AW area loses heat to the atmosphere in an annual mean. Five different products of surface heat flux are used: ERA-Interim (reanalysis), JRA-55 (reanalysis), NCEP/NCAR (reanalysis), NOC (in situ observations) and J-OFURO3 (satellite observations). See also Fig. 5.

| **Product** | ERA-Interim | JRA | NCEP/NCAR | NOC | J-OFURO |
|---|---|---|---|---|---|
| $< \mathbf{Q} >_1 / \mathbf{A}$ | 69 | 83 | 80 | 61 | 81 |
| $< \mathbf{Q} >_2 / \mathbf{A}$ | 68 | 82 | 82 | 68 | 74 |
| $-(< \mathbf{Q} >_1 - < \mathbf{Q} >_2)/\mathbf{A}$ | -1 | -1 | 2 | 7 | -7 |