# Peer review of "Mechanisms of decadal changes in sea surface height and heat content in the eastern Nordic Seas"

_Ocean Science, 2019_

## Referee Comment (RC1) · Anonymous Referee #1 · 3 Jan 2020

The paper investigates the variability of the of ssh and heat content in the "Atlantic" (Eastern) part of the Nordic Seas, the main connection of heat and salt to the Arctic Ocean. A main conclusion is that the decadal variability in this Atlantic domain can be explained by a model solely forced with the Atlantic inflow temperature variability and setting the time scale of the considered volume to some years. While this result might not appear surprising, the scaling discussion of the relative influence of the volume flux vs temperature effect on the heat content and ssh is interesting, novel and warrants publication. Also, the paper is well written and easy to follow. With this overall positive impression, there are some points listed below that I hope the author will consider.

[Figure]

The author should clearly state what time scales these results applies to. In the data and method section they should describe how the data are processed before going into the analysis. I find some information for the ssh and hydrography data (for atmospheric data little information is given). It is necessary to provide more details on this, e.g how are the data de-seasoned, are the results sensitive to the choice of method.

A main conclusion of the paper is that a simplified model of ocean heat convergence, with only upstream temperature measurements at the inflow to the Nordic Seas as input, is able to reproduce key aspects of the decadal variability of the Nordic Seas. The authors briefly mention that the residual could be related to changes in vertical heat flux or volume flux, but none of these are investigated. Their argument based on the decadal comparison of surface fluxes to exclude the heat fluxes in their forward model integration seems weakly justified. Adding to this, Mork et al. (2015) concluded that air-sea heat fluxes explained about half of the interannual (year-to-year) variability in heat content tendency. Further, from the hydrographic data the authors could check their assumption of a similar AW temperature and outflow temperature. Since a number of papers already have pointed to the importance of temperature anomalies from the North Atlantic propagating through the Norwegian Sea, I think more conclusive results on what mechanism explaining the residual variability (i.e. not explained by inflow-T) would make the paper more novel. E.g. by extending the model to include some of the above points?

Third, the authors find that the correlation between the ssh and heat content is low (provide number). The authors have used a fixed depth 657m to calculate the heat content. However, as the Atlantic Layer in the Lofoten Basin extend deeper than this, and is time-varying, the authors should assure that their results are not a sensitive to the choice of heat content integration depth. This could be tested calculating the heat content down to e.g. 1000m. Also, regarding the interpretation of the baroclinic transport function (Fig2b) as a strengthening of the Baroclinic Front Branch at the expense of the Slope Branch. It seems that the positive anomaly is quite far from the slope. The

core of this anomaly seems to match the Lofoten Eddy (that varies both in strength and position). Can you exclude that this is not the signature of the Lofoten Eddy that is smoothed in the hydrographic data set?

Page 9. Regarding the connection to the upstream North Atlantic the authors interpret this as a disconnection between the North Atlantic and the Nordic Seas after 2005. The authors could also consider the interpretation that they follow with the Nordic Seas lagging the North Atlantic. That mean that the Nordic Seas in the year to come would experience a decreasing inflow temperature and subsequent decrease in ocean heat content. Please consider this.

Minor comments:

The authors should limit the use of phrases as "key aspects", "definite similitude" without providing any statistical measure. When possible please quantify, and also preferable also include the effective number of freedom when claiming significance (e.g. Fig. 3).

Clearly describe how the data are filtered before going into further analyses, how is annual cycle removed, in what way are results sensitive to methods.

Page 8, line 23-24: Asbjørnesen et al (2018) used a volume-mean time-evolving temperature as reference.

In conclusion the authors repeat the main result both in the second paragraph and then "in the main findings". Once is better.

Conclusion two last paragraphs: I like the the point about the implication for the downstream Atlantification. However, most of the remainder should go into introduction or discussion.

In general, when there are not strong arguments against it figures should show the same area.

Fig 1. Consider including depth contours on the map.

Figure 2b. This is probably not potential energy, but more a Baroclinic Transport function. Change the title of figure. Define Sverdrup, Sv

Figure 4. The time axes are different for a and b.

Figure 6. Why do you use a 24-month running mean here?

Figure 8. The figure caption is not clear. Assure clarity.

―――――――――――――――――――――――

---

## Referee Comment (RC2) · Anonymous Referee #2 · 7 Jan 2020

Overall assessment

This manuscript treats variations of ocean heat content and sea surface height in the Atlantic domain of the Nordic Seas. It has clear illustrations and includes thorough analyses of the data sets presented as well as a relevant conceptual model. I therefore feel that it has the potential to become an important addition to the literature on the Nordic Seas. There are, however, parts of the manuscript that seem weak. I therefore feel that major revisions are required before the manuscript can be accepted for publication in Ocean Science. Below, I first address my two main concerns with the manuscript and then list some details.

[Figure]

The causal link between ocean heat content and sea level height

The manuscript links changes in sea surface height, SSH, to changes in ocean heat content below each square meter, H, i.e. to steric height changes. This seems to be one of the main conclusions of the manuscript and is stated explicitly several times:

1. "the trend in SSH is to a first approximation caused by a uniform warming of the AW" (page 4, line 19).

2. "the steric height changes related to the variation in heat content is the main reason for the observed decadal changes in SSH trends" (page 10, line 31-32).

3. "the most plausible cause of changes in SSH and heat content decadal trends is a change of temperature of the Atlantic source waters entering the Nordic Seas over the Greenland–Scotland Ridge" (page 11, line 3-4).

4. "the main reason for the shift in decadal trends in the SSH is the steric height changes related to heat content." (page 11, line 11-12).

That warming of ocean water causes expansion and thus increasing steric height is a well established fact, as long as salinity changes do not compensate too much. In the Atlantic water entering the Norwegian Sea, salinity variations have usually been parallel to temperature variations. So, there is compensation, but only partial. Thus, a warming of the Atlantic water is expected to give increased steric height. There is nothing new in that, so this cannot be one of the main conclusions of the manuscript. But, what then are the authors claiming? In spite of the many statements of this causal link listed above, it is not clear to me more precisely what they are claiming and how they justify their claim. The only justification I find for claiming that steric height changes (i.e., expansions/contractions) are the main cause of the SSH changes is Figure 4 and the discussion on it. This figure does show a qualitative correspondence between SSH and H for the defined domain, for the period 1993-2002 (although not really after that or on shorter time scales). To claim that steric height changes are the "main" cause

of the SSH changes needs a quantitative justification as well, however. I therefore find it strange that there is no calculation of the steric height changes associated with the heat content changes. This should be easy to calculate from their hydrographic data set. Why does the right panel in Figure 2 show potential energy rather than steric height. It may well be that the potential energy "largely mirrors the trend in steric height (not shown)" (page 4, line 12), but this choice makes it difficult to make a quantitative comparison between the two panels in Figure 2 and verify that steric height changes are the "main" cause of the SSH changes. Personally, I doubt that there is a quantitative justification for this claim. Using the two trend lines for the 1993-2002 period in Figure 4, the ratio between SSH change and H change is: $\Delta SSH/\Delta H \approx 4 \cdot 10\text{-}11$ m3 J-1. I don't have the hydrographic data set used by Broomé et al., but using CTD data from a standard section in the Faroe-Shetland Channel, I found a high correlation (R > 0.97) between $\Delta SSH$ calculated as steric height and $\Delta H$, but regression analyses gave $\Delta SSH/\Delta H < 2 \cdot 10\text{-}11$ m3 J-1, i.e. only around half of that in Figure 4 or less. For a vertically homogeneous water column, it is easily seen that the ratio between steric height and energy changes is $\Delta SSH/\Delta H \approx \alpha/cp$ where $\alpha$ is the isobaric expansivity and cp the specific heat per volume. Since $\alpha$ increases strongly with temperature, a ratio as high as implied in Figure 4, requires considerably warmer water than generally found in the (depth averaged) specified domain. But, more fundamentally: If Figure 4 is the justification, then the authors must imply that SSH changes in the specified domain are mainly caused by expansions/contractions within this domain. Why link SSH in the region to heat content in the region, otherwise? As argued above, there is some (not overwhelming) qualitative support for that but no quantitative justification. I doubt, however, that this can be their claim. Most of the water that was within the domain in 2002, was outside it in 1993 (probably west of the Iceland-Scotland Ridge). Thus, much of the expansion caused by warming from 1993 to 2002 will have occurred outside of the domain, perhaps in the southeastern boundary of the SPNA. This interpretation would be consistent with the statement in bullet point 3 above but, if they are really claiming that the SSH changes in the specified domain are mainly caused by

expansions/-contractions upstream of the domain, then why use the local heat content in the domain (Figure 4)? Why not discuss heat content over a wider region upstream of the domain (which would be warmer and therefore have a $\Delta$SSH/$\Delta$H ratio more consistent with Figure 4)? But, then it would of course also be necessary to evaluate the effect of circulation changes (e.g., subpolar gyre). It is well known that steric effects (thermal expansion) are an important component of recent global sea level rise (e.g. IPCC). That does not imply that the warming in a small region, as the one treated here, is the main cause of sea level rise in that region as apparently claimed. As argued above, the results presented in this manuscript rather imply the opposite. One might argue that this is a question of semantics. As defined by Eq. (1), the steric height is a mathematical construct with a value depending on the reference density, Eq. (3). Establishing a mathematical relationship with another parameter (ocean heat in the specified region) is of course fully justified. The problem arises when words like "mechanism", "cause", and "reason" are used because they imply a causal physical relationship. From a physical point of view, the statement "the main reason for the shift in decadal trends in the SSH is the steric height changes related to heat content" (page 11, line 11-12) must mean: "the main reason for the shift in decadal trends in the SSH is the expansion/Âňcontraction due to temperature changes". When SSH (which is a physical parameter; not a mathematical construct) is linked to steric height, it is linked to the physical mechanism of expansion/contraction and it has to be clearly stated where this mechanism operates. And justified based on that. The question of steric height variation in the Nordic Seas has been addressed by various authors as referred to in the manuscript. Nevertheless, I feel that the data presented in this manuscript may contribute to this topic. For that purpose, the authors need, however, to be more precise. If they want to maintain a strong causal link between steric height and SSH, they need to specify where the associated expansions/Âňcontractions have occurred and they must justify their claim quantitatively as well as qualitatively.

The conceptual model

The conceptual model in Sect. 3.2 is an appropriate component of the manuscript and helps justify the three last main findings as summarized on page 11. It raises a few questions, however:

Firstly, why use 700 m for the depth of the AW here (page 9, line 6), when 657 m is used elsewhere in the manuscript ?

Secondly, the last part of Eq. (14) defines $\tau$ in terms of the volume inside the chosen Atlantic domain, which you must have calculated (from Figure 1 and the arguments on page 8, line 10-12, it seems to be $\approx 5 \cdot 1011$ m2 $\cdot$ 657 m) and the volume transport. Using 5 Sv (page 8, line 10), this gives $\tau \approx 2$ years. I understand why you chose to use higher values, but it might be appropriate to include a sentence or two to justify this.

Thirdly – and most importantly – the arguments on page 8 for neglecting transport variations relative to temperature variations seem weak. With the uncertainties involved, the ratio 0.3/0.4 is hardly different from 1. Also, it would have been more appropriate to consider the ratio between the two driving terms in Eq. (12) rather than in Eq. (14). Then Eq. (15) would have $\Delta T'$ instead of Ti', which I assume would make the ratio closer to (or above ?) unity. To utilize this, you would, of course, need time series of volume transport in addition to temperature. From page 8, lines 28-30, you might already have this available from altimetry, but, if not, Figure 10 in Østerhus et al. (2019) provides a time series of Atlantic inflow to the Arctic Mediterranean and most of that enters between Iceland and Scotland i.e. into your Atlantic domain (Figure 9 in Østerhus et al. (2019)). As stated in your manuscript (page 8, line 27-28) this transport is highly stable on decadal time scales, but the observations do indicate an increase of at least 0.5 Sv from the mid-1990s to the early 2000s, i.e. in the period where you observe the largest increase in heat content. A back-of-the-envelope calculation indicates that including such an increase (followed by constant transport or the time series in Østerhus et al. (2019)) might give a considerably better fit than the one seen in the lower panel of Figure 6. In connection with this, the two sentences "Equation (14) is based on the reasonable assumption that the low-frequency ocean heat convergence

is dominated by changes of the AW circulation" (page 8, line 4-5) and "...variations in temperature are slightly more important than variations in volume flow" (page 8, line 14) seem contradictory.

Details

Including both "time-varying" and "trends" in the title seems a bit of an overkill and makes the title ambiguous. Do you mean "time-varying trends" ? Perhaps rephrase the title.

In the text, the Nordic Seas are sometimes treated as plural (are/have) and sometimes as singular (is/has). I prefer plural, but in any case, choose one.

Page 1, line 5: can "slowdown" -> "weakening"

Page 1, line 21: "Chafik and Rossby (2019)" -> "(Chafik and Rossby, 2019)"

Page 2, line 8: "have" -> "has"

Page 2, line 13: "carry" -> "carries"

Page 2, line 24: "stagnant" -> "slowly-increasing"

Page 2, line 26: "variations" -> "variation"

Page 2, line 32-33: Do you actually use "Absolute" (rather than SLA) altimetry data ?

Page 3, line 6: Non-standard reference

Page 4, line 2: "have" -> "has"

Page 4, line 8: "dynamic sea surface height" -> "sea surface height"

Page 4, line 22: "extent that" -> "extent than that"

Page 4, line 29: "mean flow heat transport" ?????????

Page 5, line 17: "seasonal variation in heat content" -> "seasonal variation in heat

content and wind forcing"

Page 6, line 19: "average" -> "averaged"

Page 6, line 29: "show" -> "shows"

Page 7, line 1: "data is" -> "data are"

Page 7, line 13-17: Defining the overbar parameters as "time-mean" (line 13) is inconsistent with Eq. (11) (line 15) before you neglect second order terms (line 17). I suggest to move this assumption up. Then Eq. (11) follows naturally and is not a "choice".

Page 11, line 18: "seem" -> "seems"

Page 11, line 18: "maintain" -> "maintains"

Page 14, line 16: Non-standard reference

Figure 3 and Figure 8: It is nowhere stated, how statistical significance is estimated, specifically whether it takes serial correlation into account. If it does take this into account, this should be stated (e.g. in the figure captions). If it does not, the significance should be re-calculated and the figure modified or the dots in Figure 3 and circles in Figure 8 should be removed as well as any reference to statistical significance in the captions and text.

The two panels in Figure 6 are labeled a) and b). In other two-panel figures, you use left/right or upper/lower. Be consistent.

---

## Author Response (AR1)

1. Comments from referees
2. Authors response
3. Additional comments on changes in manuscript with page-reference to diff-file

Answers to review #1

The paper investigates the variability of the of ssh and heat content in the "Atlantic" (Eastern) part of the Nordic Seas, the main connection of heat and salt to the Arctic Ocean. A main conclusion is that the decadal variability in this Atlantic domain can be explained by a model solely forced with the Atlantic inflow temperature variability and setting the time scale of the considered volume to some years. While this result might not appear surprising, the scaling discussion of the relative influence of the volume flux vs temperature effect on the heat content and ssh is interesting, novel and warrants publication. Also, the paper is well written and easy to follow. With this overall positive impression, there are some points listed below that I hope the author will consider.

We thank the reviewer for the constructive comments and careful inspection of our manuscript and results. The insightful comments by the reviewer have been taken into account. We also note (as the reviewer points out) that we have been brief at some places and we have now made an effort to include the necessary details. The replies to each of the comments by the reviewer can be found in the attached pdf below in blue.

The author should clearly state what time scales these results applies to. In the data and method section they should describe how the data are processed before going into the analysis. I find some information for the ssh and hydrography data (for atmospheric data little information is given). It is necessary to provide more details on this, e.g how are the data de-seasoned, are the results sensitive to the choice of method.

We thank the reviewer for pointing this out. Additions to clearly point out the time scales considered, in both Section 1 and 4, will make the paper easier to grasp and also express novelty. See also further comments below. We now also extend Section 2 to include more information on how the ssh, hydrographic and air-sea heat flux data have been processed before the analysis.

To clearly point out what time scales the analysis concerns, the title has been changed to include the word "decadal". The first two paragraphs of Section 4 have also been revised to strengthen this message.
Additions to the data processing have been added at P3 L8-9, P4 L19-20.

A main conclusion of the paper is that a simplified model of ocean heat convergence, with only upstream temperature measurements at the inflow to the Nordic Seas as input, is able to reproduce key aspects of the decadal variability of the Nordic Seas. The authors briefly mention that the residual could be related to changes in vertical heat flux or volume flux, but none of these are investigated. Their argument based on the decadal comparison of surface fluxes to exclude the heat fluxes in their forward model integration seems weakly justified. Adding to this, Mork et al. (2015) concluded that air-sea heat fluxes explained about half of the interannual (year-to-year) variability in heat content tendency. Further, from the hydrographic data the authors could check their assumption of a similar AW temperature and outflow temperature. Since a number of papers already have pointed to the importance of temperature anomalies from the North Atlantic propagating through the Norwegian Sea, I think more conclusive results on what mechanism explaining the residual variability (i.e. not explained by inflow-T) would make the paper more novel. E.g. by extending the model to include some of the above points?

Answer: We thank the Reviewer for these comments. The decadal comparison of surface fluxes essentially rules out the role for air-sea fluxes for driving the heat content anomalies. This comparison should not be compared to the study by Mork et al. (2015) because of the different time scales, interannual vs. decadal. The latter is the focus of our study, where the decadal comparison shows a minor role for air-sea fluxes, which further suggests that on these time scales advection of temperature anomalies by ocean currents from the North Atlantic dominates the heat budget. The decadal time scales considered also adds to the novelty.

We have decided to not explore how well the mean AW temperature is related to the outflow temperature, defined at the northern boundary of our domain located roughly in the Lofoten Basin. For variations with time scales of a few years this should be a reasonable assumption, since at low frequencies the temperature anomalies in the AW tend to have large horizontal scales.

In response to questions from both referees on the residual in the conceptual model, we have expanded the discussion on the relative importance of anomalies in volume flow and temperature. We have changed some text on page 8 and 9: We now state that observations of Mork and Skagseth (2010), Berx et al. (2013), Bringedal et al. (2018), and Østerhus et al. (2019) suggest an increase of the Atlantic inflow to the Nordic Seas from the mid 1990:s to the early 2000:s. Further, we now write that this increase in volume transport can qualitatively explain why the observed temperatures are higher

than those in the conceptual model (which is forced only by temperature variations) in the early 2000:s.

Regarding the residual variability, Fig. 6 now also includes the temperature anomaly forcing needed to reproduce the observed heat content (see P10 L11-14, P10 L23-25 and Fig 6) and the possible relation between the volume inflow and the residual is discussed (P11 L1-6).

Third, the authors find that the correlation between the ssh and heat content is low (provide number). The authors have used a fixed depth 657m to calculate the heat content. However, as the Atlantic Layer in the Lofoten Basin extend deeper than this, and is time-varying, the authors should assure that their results are not a sensitive to the choice of heat content integration depth. This could be tested calculating the heat content down to e.g. 1000m. Also, regarding the interpretation of the baroclinic transport function (Fig2b) as a strengthening of the Baroclinic Front Branch at the expense of the Slope Branch. It seems that the positive anomaly is quite far from the slope. The core of this anomaly seems to match the Lofoten Eddy (that varies both in strength and position). Can you exclude that this is not the signature of the Lofoten Eddy that is smoothed in the hydrographic data set?

Answer: We thank the author for this suggestion concerning the sensitivity to the integration depth. We already had a sentence about the integration depth (end of Section 2.2), but this only concerned the spatial pattern of the heat content, and not the correlation with the SSH. After testing the correlation to the heat content integrated to 1000m, it does not seem to be sensitive to the choice of depth. We have now extended the discussion on the sensitivity to integration depth.

About the signature of the Lofoten Eddy: the largest trend of the potential energy (as well as the steric height, now shown in the revised Fig. 2) in the central part of the Lofoten Basin indeed coincides with where the Lofoten Vortex resides. As suggested by the referee, the larger-scale positive trend seen in potential energy, steric height, and heat content (Skagseth and Mork, 2012) in time- and spatially-averaged hydrographic data sets may reflect an increase in intensity and number of mesoscale anticyclonic eddies in the Lofoten Basin (Köhl, 2007, Raj et al., 2015). These eddies have warm cores and locally yield positive anomalies in heat content, steric height and potential energy. Strengthening of the intensity of a generally dominating anticyclonic eddy, known as the Lofoten Eddy or Vortex (Søiland et al., 2016) can be important, but this appears to be linked to periods with a higher number of anticyclonic eddies entering the

Lofoten Basin from continental slope. To point to this slightly different perspective of warming trend in the Lofoten Basin, we have added at the end of section (3.1):

"The broad-scale positive trend in heat content, steric height and potential energy in the Lofoten Basin, recorded in the time-averaged and space-interpolated hydrographic data, may reflect an increase in intensity and number of mesoscale anticyclonic eddies shed from the continental slope that propagate into the central basin (Köhl, 2007, Raj et al., 2015, Chafik et al. 2015). Higher influx of eddies from the slope can invigorate the long-lived dominating anticyclonic eddy (Köhl, 2007), known as the Lofoten Vortex, which has a strong local hydrographic signature and moves around in the central basin (Søiland et al., 2016). However, for the time-mean flow trend it does not matter if the warming in the Lofoten Basin is due to meso-scale eddy-induced or large-scale induced heat convergence."

The mention of the correlation between the monthly time series has been removed, since that is not the relevant time scale for this work and might only confuse the reader. The last part of Section 2.2, about the sensitivity to integration depth, has been slightly extended.
The paragraph added, as seen above, has been combined with some material moved out from Section 4 and now reads (see last paragraph in Section 3.1):

"The broad-scale positive trend in heat content, steric height and baroclinic transport function in the Lofoten Basin, recorded in the time- and space-interpolated hydrographic data, may partly reflect an increase in the intensity and number of mesoscale anticyclonic eddies shed from the continental slope that propagate into the central basin (Köhl, 2007; Raj et al., 2015; Chafik et al., 2015). Higher influx of eddies from the slope can invigorate the long-lived dominating anticyclonic eddy (Köhl, 2007), known as the Lofoten Vortex, which has a strong local hydrographic signature and moves around in the central basin (Søiland et al., 2016). The associated changes in steric height (Fig. 2) in the Lofoten Basin have served to induce an anticyclonic flow anomaly carrying a larger fraction of AW from the slope current into the basin. This flow anomaly acts to enhance the near-surface heat transport by the mean flow entering the Lofoten Basin from south (Dugstad et al., 2019). In combination with alterations of eddy fluxes from the Lofoten escarpment (Spall, 2010; Chafik et al., 2015) the anticyclonic mean-flow anomaly is a plausible mechanism for the build-up of the Lofoten Basin heat content over the period 1993-2017. However, for the large-scale trend pattern it does not matter if the warming in the Lofoten Basin is caused by mesoscale eddies or by mean-flow changes."

Page 9. Regarding the connection to the upstream North Atlantic the authors interpret this as a disconnection between the North Atlantic and the Nordic Seas after 2005. The authors could also consider the interpretation that they follow with the Nordic Seas lagging the North Atlantic. That mean that the Nordic Seas in the year to come would experience a decreasing inflow temperature and subsequent decrease in ocean heat content. Please consider this.

Answer: We thank the Reviewer for this comment. Since our data stopped in 2016, we chose not to speculate about a possible lag. Instead, we focused on the possible mechanisms for this disconnection. However, we realize that doing as the reviewers suggests could have helped the interpretation of the disconnection. Analysis (not shown) of the recent years in fact indeed hints on a lag mechanism from the North Atlantic. We have now commented on this connection in the manuscript. P12 L8-9.

Minor comments:

The authors should limit the use of phrases as "key aspects", "definite similitude" without providing any statistical measure. When possible please quantify, and also preferable also include the effective number of freedom when claiming significance (e.g. Fig. 3).

Answer: We thank the reviewer for careful reading of our manuscript and take this into consideration in the revised version. The methods used to show significance are also added in the new version. See captions of figures 3 and 8.

Clearly describe how the data are filtered before going into further analyses, how is annual cycle removed, in what way are results sensitive to methods.

Answer: This is extended in Section 2.

Page 8, line 23-24: Asbjørnesen et al (2018) used a volume-mean time-evolving temperature as reference.

Answer: It is true that Asbjørnsen et al. (2018) used a volume-mean time-varying temperature of the Norwegian Sea. However, they concluded that they got essentially the same results if they used 0 °C as a fixed reference temperature, which was used by Orvik and Skagseth (2005). To avoid possible misunderstandings we now instead cite Orvik and Skagseth (2005) for the reference temperature.

In conclusion the authors repeat the main result both in the second paragraph and then "in the main findings". Once is better.

Answer: The Conclusions are revised. The list of "main findings" is removed, and the last item is moved to be included in the second paragraph. This paragraph will also be modified to better clarify the proposed answer to the question on remote or local forcing.

Conclusion two last paragraphs: I like the the point about the implication for the downstream Atlantification. However, most of the remainder should go into introduction or discussion.

Answer: The paragraph starting at L20 is moved into section 3.1, to the discussion at the end where we have also added a paragraph on the Lofoten vortex (see above).

See P5 L26-30

In general, when there are not strong arguments against it figures should show the same area.
Answer: Figures 1-3 and 5 are revised to show the same area.

Figures 2, 3 and 5 now show the same area. Figure 1 is kept with a little bit of a wider view, for orientation.

Fig 1. Consider including depth contours on the map.
Answer: Done.

Figure 2b. This is probably not potential energy, but more a Baroclinic Transport function. Change the title of figure. Define Sverdrup, Sv

Answer: The content of Figure 2 is changed and clarified. Figure 2 has been improved by also adding steric height trends. We now define Sv.

Figure 4. The time axes are different for a and b.

Answer: That is correct and was intentional. The hydrographic data used only covers the period from 1993-2016, and we still wanted to show the full altimetry data available. This could have been clearly stated in the figure caption. In the new version, the figure is modified so that the time axes are the same, but with the last data points empty for the hydrographic data, to avoid confusion.

Figure 6. Why do you use a 24-month running mean here?

Answer: A 24-month running is used to reflect inter-decadal variability; the focus of this study. For consistency, the same filter length is now also used in Fig. 4.

Figure 8. The figure caption is not clear. Assure clarity.

Answer: The caption is now revised. We thank the Reviewer for careful inspection.

**Answers to review #2**

Overall assessment

This manuscript treats variations of ocean heat content and sea surface height in the Atlantic domain of the Nordic Seas. It has clear illustrations and includes thorough analyses of the data sets presented as well as a relevant conceptual model. I therefore feel that it has the potential to become an important addition to the literature on the Nordic Seas. There are, however, parts of the manuscript that seem weak. I therefore feel that major revisions are required before the manuscript can be accepted for publication in Ocean Science. Below, I first address my two main concerns with the manuscript and then list some details.

We wish to thank the reviewer for the many constructive comments and for carefully reading our manuscript. We have, to the best of our knowledge, addressed all major and minor comments (see blue text below) . Doing so, we think that the comments altogether have helped to produce an improved version of our manuscript.

The causal link between ocean heat content and sea level height

The manuscript links changes in sea surface height, SSH, to changes in ocean heat content below each square meter, H, i.e. to steric height changes. This seems to be one of the main conclusions of the manuscript and is stated explicitly several times:

1. "the trend in SSH is to a first approximation caused by a uniform warming of the AW" (page 4, line 19).

2. "the steric height changes related to the variation in heat content is the main reason for the observed decadal changes in SSH trends" (page 10, line 31-32).

3. "the most plausible cause of changes in SSH and heat content decadal trends is a change of temperature of the Atlantic source waters entering the Nordic Seas over the Greenland–Scotland Ridge" (page 11, line 3-4).

4. "the main reason for the shift in decadal trends in the SSH is the steric height changes related to heat content." (page 11, line 11-12).

That warming of ocean water causes expansion and thus increasing steric height is a well established fact, as long as salinity changes do not compensate too much. In the Atlantic water entering the Norwegian Sea, salinity variations have usually been parallel to temperature variations. So, there is compensation, but only partial. Thus, a warming

of the Atlantic water is expected to give increased steric height. There is noth- ing new in that, so this cannot be one of the main conclusions of the manuscript. But, what then are the authors claiming? In spite of the many statements of this causal link listed above, it is not clear to me more precisely what they are claiming and how they justify their claim. The only justification I find for claiming that steric height changes (i.e., expansions/contractions) are the main cause of the SSH changes is Figure 4 and the discussion on it. This figure does show a qualitative correspondence between SSH and H for the defined domain, for the period 1993-2002 (although not really after that or on shorter time scales). To claim that steric height changes are the "main" cause of the SSH changes needs a quantitative justification as well, however. I therefore find it strange that there is no calculation of the steric height changes associated with the heat content changes. This should be easy to calculate from their hydrographic data set. Why does the right panel in Figure 2 show potential energy rather than steric height. It may well be that the potential energy "largely mirrors the trend in steric height (not shown)" (page 4, line 12), but this choice makes it difficult to make a quantitative comparison between the two panels in Figure 2 and verify that steric height changes are the "main" cause of the SSH changes. Personally, I doubt that there is a quanti- tative justification for this claim. Using the two trend lines for the 1993-2002 period in Figure 4, the ratio between SSH change and H change is: $\Delta SSH/\Delta H \approx 4 \cdot 10^{-11}$ m3 J-1. I don't have the hydrographic data set used by Broomé et al., but using CTD data from a standard section in the Faroe-Shetland Channel, I found a high correlation (R > 0.97) between $\Delta SSH$ calculated as steric height and $\Delta H$, but regression analyses gave $\Delta SSH/\Delta H < 2 \cdot 10^{-11}$ m3 J-1, i.e. only around half of that in Figure 4 or less. For a vertically homogeneous water column, it is easily seen that the ratio between steric height and energy changes is $\Delta SSH/\Delta H \approx \alpha/cp$ where $\alpha$ is the isobaric expansivity and cp the specific heat per volume. Since $\alpha$ increases strongly with temperature, a ratio as high as implied in Figure 4, requires considerably warmer water than generally found in the (depth averaged) specified domain. But, more fundamentally: If Figure 4 is the justification, then the authors must imply that SSH changes in the specified domain are mainly caused by expansions/contractions within this domain. Why link SSH in the region to heat content in the region, otherwise? As argued above, there is some (not overwhelming) qualitative support for that but no quantitative justification. I doubt, however, that this can be their claim. Most of the water that was within the domain in 2002, was outside it in 1993 (probably west of the Iceland-Scotland Ridge). Thus, much of the expansion caused by warming from 1993 to 2002 will have occurred outside of the domain, perhaps in the southeastern boundary of the SPNA. This in- terpretation would be consistent with the statement in bullet point 3 above but, if they are really claiming that the SSH changes in the specified domain are mainly caused by expansions/-contractions upstream of the domain, then why use the local heat content

in the domain (Figure 4)? Why not discuss heat content over a wider region upstream of the domain (which would be warmer and therefore have a $\Delta$SSH/$\Delta$H ratio more consistent with Figure 4)? But, then it would of course also be necessary to evaluate the effect of circulation changes (e.g., subpolar gyre). It is well known that steric ef- fects (thermal expansion) are an important component of recent global sea level rise (e.g. IPCC). That does not imply that the warming in a small region, as the one treated here, is the main cause of sea level rise in that region as apparently claimed. As ar- gued above, the results presented in this manuscript rather imply the opposite. One might argue that this is a question of semantics. As defined by Eq. (1), the steric height is a mathematical construct with a value depending on the reference density, Eq. (3). Establishing a mathematical relationship with another parameter (ocean heat in the specified region) is of course fully justified. The problem arises when words like "mechanism", "cause", and "reason" are used because they imply a causal physical relationship. From a physical point of view, the statement "the main reason for the shift in decadal trends in the SSH is the steric height changes related to heat content" (page 11, line 11-12) must mean: "the main reason for the shift in decadal trends in the SSH is the expansion/Ânˇcontraction due to temperature changes". When SSH (which is a physical parameter; not a mathematical construct) is linked to steric height, it is linked to the physical mechanism of expansion/contraction and it has to be clearly stated where this mechanism operates. And justified based on that. The question of steric height variation in the Nordic Seas has been addressed by various authors as referred to in the manuscript. Nevertheless, I feel that the data presented in this manuscript may contribute to this topic. For that purpose, the authors need, however, to be more precise. If they want to maintain a strong causal link between steric height and SSH, they need to specify where the associated expansions/Ânˇcontractions have occurred and they must justify their claim quantitatively as well as qualitatively.

Answer: We thank the referee for a detailed and insightful discussion on the physical link between ocean heat content and sea level.

The reviewer is correct in pointing out that it is well established that ocean warming causes increasing (steric) sea level height. We have modified the manuscript to more clearly convey that the novel results on this subject concerns decadal time scales and the importance of advection of temperature anomalies from the North Atlantic, which appears to be dominating over local air-sea fluxes on decadal timescales. Air-sea heat fluxes are important for inter-annual ocean temperature anomalies in the eastern Nordic Seas as have been shown in several previous studies.

Points 3 and 4 in the Reviewers quotes are both changed in the new version.

The reviewer is correct in stating that the sea level cannot uniquely be divided in a steric height component and bottom pressure component as one has to select a somewhat arbitrary reference density. However, when one considers changes in the sea level, the steric changes can be tied to the observed changes in density, and this measure of steric height changes is essentially insensitive to the reference density (as long as it is taken to be a typical mean ocean value). In section 2.2, we now describe how the sea level can be partitioned into a steric height and a bottom pressure component and that changes in steric height are causally linked to the vertically-integrated changes in density. Here we now cite Gill and Niiler (1973), which describes the theoretical underpinning of these concepts.

This new and more careful derivation and discussion in section 2.2 should clarify that the steric height changes are tied to the local vertically-integrated changes in density (or buoyancy), which in turn can be due either to local air-sea fluxes or due to oceanic buoyancy convergence caused for instance by advection of upstream water with anomalous buoyancy. We believe that this answers the referee's question about the role of expansion caused by warming in the North Atlantic south of the Nordic Seas: this steric height signal is tied to the buoyancy anomaly of water advected northward.

The referee asks whether the thermal expansion due to the heat content changes can explain the altimetric sea level changes in Figure 4, and provide some back of the envelope estimates. We have now added the steric height variations calculated from the hydrographic data (EN4 data set provided by the UK Met Office) in Figure 4. This shows that the steric height and ocean heat content variations in the hydrographic data are strongly correlated (as expected) and that the steric height variations explain main parts of the inter-decadal variations in the altimetric sea level. We now also cite Chafik et al. (2019), who show that inter-decadal variation in the surface wind stress explain some of the sea level variation in the eastern Nordic Seas (P6 L20-21).

Also in Figure 2, we now show the trend in steric height calculated from the hydrographic data.

The conceptual model

The conceptual model in Sect. 3.2 is an appropriate component of the manuscript and helps justify the three last main findings as summarized on page 11. It raises a few questions, however:

Firstly, why use 700 m for the depth of the AW here (page 9, line 6), when 657 m is used elsewhere in the manuscript ?

Secondly, the last part of Eq. (14) defines τ in terms of the volume inside the chosen Atlantic domain, which you must have calculated (from Figure 1 and the arguments on page 8, line 10-12, it seems to be ≈ 5·1011 m2 · 657 m) and the volume transport. Using 5 Sv (page 8, line 10), this gives τ≈2 years. I understand why you chose to use higher values, but it might be appropriate to include a sentence or two to justify this.

Thirdly – and most importantly – the arguments on page 8 for neglecting transport variations relative to temperature variations seem weak. With the uncertainties involved, the ratio 0.3/0.4 is hardly different from 1. Also, it would have been more appropriate to consider the ratio between the two driving terms in Eq. (12) rather than in Eq. (14). Then Eq. (15) would have $\Delta T'$ instead of $T_i'$, which I assume would make the ratio closer to (or above ?) unity. To utilize this, you would, of course, need time series of volume transport in addition to temperature. From page 8, lines 28-30, you might already have this available from altimetry, but, if not, Figure 10 in Østerhus et al. (2019) provides a time series of Atlantic inflow to the Arctic Mediterranean and most of that enters between Iceland and Scotland i.e. into your Atlantic domain (Figure 9 in Øster- hus et al. (2019)). As stated in your manuscript (page 8, line 27-28) this transport is highly stable on decadal time scales, but the observations do indicate an increase of at least 0.5 Sv from the mid-1990s to the early 2000s, i.e. in the period where you observe the largest increase in heat content. A back-of-the-envelope calculation indi- cates that including such an increase (followed by constant transport or the time series in Østerhus et al. (2019)) might give a considerably better fit than the one seen in the lower panel of Figure 6. In connection with this, the two sentences "Equation (14) is based on the reasonable assumption that the low-frequency ocean heat convergence is dominated by changes of the AW circulation" (page 8, line 4-5) and ". . .variations in temperature are slightly more important than variations in volume flow" (page 8, line 14) seem contradictory.

Answer:

Conceptual model

First point: 657 m is the depth in the hydrographic data base closest to 700. We now state this. P10 L20

Second point: There is some degree of arbitrariness in defining the limits of the Atlantic Volume in the conceptual model. Taking the box indicated in Fig. 1 gives an area about $6 \times 10^{11}$ m$^2$, which with H~700 m and a transport of 5 Sv, one obtains a time scale of 2.6 ≈ 3 years. On page 8 after discussing the residence time scale we have added a sentence which clarifies that, by taking these specific number for the Atlantic Water

properties the residence time-scale defined in Eq. (14) gives value comparable to the ones estimated by for example Kozalka et al. (2013) P9 L6-8

Third point: We agree with the referee that our scaling arguments based on the conceptual model do not firmly show that temperature anomalies are dominating over volume transport anomalies for driving Atlantic Water temperature anomalies in the Nordic Seas. Instead, given the qualitative nature of scaling arguments it is reasonable to conclude that temperature- and transport-anomalies are potentially of equal importance. On P8 L19 we have therefore changed "can be more important for ocean heat convergence" to "can be equally important for ocean heat convergence". However the important point (P8 L20), which we want to make clear, is that for heat transport through a section, volume variations completely dominates over temperature variations (Asbjørnsen et al., 2018). We stress this difference between heat transport and heat convergence since it is important for rationalising the simple scenario where only forcing from temperature anomalies in the conceptual model is considered.

To give a more detailed view on the relative importance of anomalies in volume flow and temperature, we have changed some text on page 8 and 9: We now state that observations of Mork and Skagseth (2010), Berx et al. (2013), Bringedal et al. (2018) and Østerhus et al. (2019) suggest an increase of the Atlantic inflow to the Nordic Seas from the mid 1990:s to the early 2000:s. And further, we now write that this increase in volume transport can qualitatively explain why the observed temperatures are higher than those in the conceptual model (which is forced only by temperature variations) between in the early 2000:s. P11 L1-6

Finally, we think the referee's point concerning contradictory statement in relation to Eq. (14) can be solved by writing:  "Equation (14) is based on the reasonable assumption that the low-frequency ocean heat convergence is dominated by changes of the AW circulation, *rather than air-sea heat fluxes*" P9 L10

Details

Including both "time-varying" and "trends" in the title seems a bit of an overkill and makes the title ambiguous. Do you mean "time-varying trends" ? Perhaps rephrase the title.
Answer: Yes, we do mean "time-varying trends", but did not realize this wasn't clear. The title will be changed to: "Mechanisms of the time-varying trends in sea surface height and heat content in the Nordic Seas"

After further revision, we changed the title to: "Mechanisms of decadal changes in sea surface height and heat content in the eastern Nordic Seas"

In the text, the Nordic Seas are sometimes treated as plural (are/have) and sometimes as singular (is/has). I prefer plural, but in any case, choose one.

Answer: We thank the Reviewer for pointing this out.

Page 1, line 5: can "slowdown" -> "weakening"

Answer: Done.

Page 1, line 21: "Chafik and Rossby (2019)" -> "(Chafik and Rossby, 2019)"

Answer: Done.

Page 2, line 8: "have" -> "has"

Answer: Done.

Page 2, line 13: "carry" -> "carries"

Answer: Done.

Page 2, line 24: "stagnant" -> "slowly-increasing"

Answer: Done.

Page 2, line 26: "variations" -> "variation"

Answer: Done.

Page 2, line 32-33: Do you actually use "Absolute" (rather than SLA) altimetry data ?
Answer: Yes.

Page 3, line 6: Non-standard reference

Answer: Fixed.

Page 4, line 2: "have" -> "has"

Answer: Done.

Page 4, line 8: "dynamic sea surface height" -> "sea surface height"

Answer: OK.

Page 4, line 22: "extent that" -> "extent than that"

Answer: No, this is supposed to be what it is. Reformulated to avoid confusion P5 L9

Page 4, line 29: "mean flow heat transport" ?????????
Answer: Changed to: "Heat transport by the mean flow" P5 L16

Page 5, line 17: "seasonal variation in heat content" -> "seasonal variation in heat content and wind forcing"

Answer: OK.

Page 6, line 19: "average" -> "averaged"

Answer: Done.

Page 6, line 29: "show" -> "shows"

Answer: Done

Page 7, line 1: "data is" -> "data are"

Answer: Done.

Page 7, line 13-17: Defining the overbar parameters as "time-mean" (line 13) is inconsistent with Eq. (11) (line 15) before you neglect second order terms (line 17). I suggest to move this assumption up. Then Eq. (11) follows naturally and is not a "choice".

Answer: This is adjusted.

Page 11, line 18: "seem" -> "seems"

Answer: Done.

Page 11, line 18: "maintain" -> "maintains"

Answer: Done.

Page 14, line 16: Non-standard reference

Answer: Fixed.

Figure 3 and Figure 8: It is nowhere stated, how statistical significance is estimated, specifically whether it takes serial correlation into account. If it does take this into account, this should be stated (e.g. in the figure captions). If it does not, the significance should be re-calculated and the figure modified or the dots in Figure 3 and circles in Figure 8 should be removed as well as any reference to statistical significance in the captions and text.

Answer: For Fig. 3, two-sided p-values are calculated using the Wald test, which is provided by the SciPy linregress routine (Oliphant, 2007*). The significance test in Fig. 8 will be revised to take into account the effective number of degrees of freedom.

*Oliphant, T. E. (2007). Python for scientific computing. Computing in Science & Engineering, 9(3), 10–20. doi: 10.1109/MCSE.2007.58

See figure captions for Fig. 3 and 8.

The two panels in Figure 6 are labeled a) and b). In other two-panel figures, you use left/right or upper/lower. Be consistent.

Answer: We thank the reviewer for this. Done.

[revised manuscript text omitted]